# Cereulide and Emetic *Bacillus cereus*: Characterizations, Impacts and Public Precautions

**DOI:** 10.3390/foods12040833

**Published:** 2023-02-15

**Authors:** Shuo Yang, Yating Wang, Yangtai Liu, Kai Jia, Zhen Zhang, Qingli Dong

**Affiliations:** School of Health Science and Engineering, University of Shanghai for Science and Technology, Jungong Road No. 334, Yangpu District, Shanghai 200093, China

**Keywords:** *Bacillus cereus*, emetic toxin, cereulide, contamination, toxicity, precautions

## Abstract

Cereulide, which can be produced by *Bacillus cereus*, is strongly associated with emetic-type food poisoning outbreaks. It is an extremely stable emetic toxin, which is unlikely to be inactivated by food processing. Considering the high toxicity of cereulide, its related hazards raise public concerns. A better understanding of the impact of *B. cereus* and cereulide is urgently needed to prevent contamination and toxin production, thereby protecting public health. Over the last decade, a wide range of research has been conducted regarding *B. cereus* and cereulide. Despite this, summarized information highlighting precautions at the public level involving the food industry, consumers and regulators is lacking. Therefore, the aim of the current review is to summarize the available data describing the characterizations and impacts of emetic *B. cereus* and cereulide; based on this information, precautions at the public level are proposed.

## 1. Introduction 

*Bacillus (B.) cereus sensu lato (s.l.)* is a genetically similar group of Gram-positive, spore-forming bacteria, which commonly play a role in agriculture, environment, food spoilage and human and animal health [1]. *B. cereus s.l.* contains diverse species, which are divided into different phylogenetic groups (I–VII) based on their genetic characteristics [2]. For instance, *B. weihenstephanensis* strains are part of group VI, while mesophilic *B. cereus sensu stricto* (hereafter referred to as *B. cereus*) strains are clustered in group III. Strains that produce enterotoxins are found in several phylogenetic groups [2]. Among these strains, *B. cereus* is the one most commonly related to foodborne illnesses [3]. According to the European Food Safety Authority (EFSA) annual report, 16–20% of food poisoning outbreaks caused by bacterial toxins are attributed to *B. cereus*. Over the period from 2011 to 2015, 220–291 food poisoning outbreaks associated with *B. cereus* were reported in several member states, accounting for 3.9–5.5% of all food poisoning outbreaks [4,5,6,7,8,9,10]. For the period of 2010–2016, 86% of the outbreaks were associated with enterotoxins produced by *Clostridium*, *Staphylococcus* and *B. cereus* [4]. In France, foodborne outbreaks caused by *B. cereus* are currently the second most common after those caused by *Staphylococcus aureus* (*S. aureus*) [11]. Moreover, more than a million food-associated illnesses occur each year due to bacterial toxins, including *B. cereus*, within the United States [12,13,14]. In 2013, the Health and Safety Executive (HSE) classified *B. cereus* as a risk group 2 (RG2) human pathogen; RG2 pathogens can cause human diseases and may be hazardous to employees [15].

*B. cereus* is ubiquitous in the environment, where it can be isolated from soil and water [16]. It can also be isolated from diverse food matrices, including cereals, rice, milk, vegetables, fruits, poultry and drinks [17]. Moreover, *B. cereus* has shown strong resistance to extreme environmental conditions due to its production of spores, which can overcome these difficulties; in particular, its spores are highly tolerant to heat, freezing, drying and UV radiation [18,19]. Once conditions are ideal, spore germination will occur and eventually lead to the outgrowth of vegetative cells. In addition, the vegetative cells subsequently form a biofilm, which protects the vegetative cells and results in an enhanced ability of the bacteria to endure extreme environmental conditions [20].

The public health concerns related to *B. cereus* are associated with the production of the emesis-inducing toxin cereulide and diarrheal-inducing toxins (HaemolysinBL, non-hemolytic enterotoxin and cytotoxin K), which causes foodborne illness [21]. The diarrheal form of *B. cereus* has an onset period of 8–16 h, while the emetic form has an onset period of 0–5 h [22]. Cereulide acts on mitochondria, leading to dysfunction in different organs (liver, pancreatic islet, brain, intestines, etc.) and body systems (immune system and nervous system) [23,24,25,26,27,28,29,30]. Cereulide is produced in the food prior to consumption and is unlikely to be inactivated during food processing, given that it is extremely stable under heat treatment at 121 °C for 2 h, roasting, frying and microwave cooking [31]. It is also resistant to exposure to a wide range of pH values (2–10) [32]. Unlike cereulide, diarrheal enterotoxins are sensitive to heat, acids or proteases and are considered to be unstable in these environments [21]. Thus, cereulide toxin is of particular concern, and once the cereulide toxin is produced in food, it cannot be eliminated during food processing and causes potential risks for consumers as a result.

The potential risks of *B. cereus* in terms of causing emetic food poisoning are partly due to the unavoidable presence of cereulide-producing strains and the persistence of cereulide during processing. Therefore, risk management should mainly focus on precautions that prevent *B. cereus* growth and cereulide production. In this review, a comprehensive overview of the current knowledge on the natural niches of emetic *B. cereus* and its prevalence in food, the toxicology profiles of cereulide, the infective dose of *B. cereus*, the emetic dose of cereulide, guideline levels for *B. cereus* and factors influencing cereulide formation is provided to obtain better insights into *B. cereus* and toxin characterization. The summarized information further serves as a basis for the creation of precautions at the public level, for which it is crucial to gain valuable input from the food industry, consumers and regulators involved.

## 2. Natural Niches and Prevalence of Emetic *B. cereus*

*B. cereus* is ubiquitous and widespread in the environment [33]. The spores and cells of *B. cereus* are commonly found in soil, water and the gastrointestinal (GI) tract of eukaryotes [34]. *B. cereus* strains are often present in the roots and tubers of plants, especially in starch-rich plants [35]. During harvest, *B. cereus* cells or spores may be transported with plant material and become established on food processing equipment, resulting in contamination at different stages of the production process, which attracts more attention in industry [36]. Contamination with *B. cereus* is widespread in foods, and its incidence is high. *B. cereus* is easily transmitted by dust and insects to contaminate food [37], and the carrier rate of food can be as high as 20–70% [38,39]. Mostafa et al. [38] screened 360 meat and milk samples and detected *B. cereus* strains in 24.4% of samples. In a study of 575 samples from food business operators, *B. cereus* was found to be present in 56% of the samples [40]. Examples of foods that see widespread contamination by *B. cereus* include cereal products, rice, seeds, dairy products, poultry, vegetables, herbs and spices, seafood, ready-to-eat products and desserts [40,41,42,43,44,45,46]. Gdoura-Ben Amor et al. [39] assessed the prevalence of *B. cereus* by analyzing 687 different samples. A total of 174 isolates contained *B. cereus*, including cereals (67.6%), pastry products (46.2%), cooked food (40.8%), poultry meat (32.7%), seafood products (32.3%), spices (28%), canned products (16.7%), raw poultry meat (9.4%), fresh-cut vegetables (5%) and dairy products (4.8%).

Although *B. cereus* is widespread, emetic *B. cereus* is rarely found in the environment. In fact, the nature niches and the ways of entry into food production and processing of emetic *B. cereus* are largely unknown [47]. Thus far, emetic *B. cereus* has been found in a wide variety of foods. For instance, emetic *B. cereus* has commonly been detected in rice. Following the culture of isolated strains from rice on agar, it was discovered that 42% of the strains produced levels of cereulide above the limit of detection, in contrast to 16.2% and 7.2% of the strains detected in farinaceous and non-farinaceous foods, respectively [48]. Moreover, in another study, emetic *B. cereus* strains were detected most commonly in pasta filata cheese, with a prevalence of 13%, followed by ready-to-eat foods (11%) and dried mushrooms (8%). In addition, a low occurrence rate was found for herbs and spices (1.7%) and seafood (1.6%) [47]. It is not difficult to notice that rice and farinaceous foods are important vehicles for *B. cereus* contamination and are involved in *B. cereus* intoxication [49]. Some other enterotoxin-producing pathogens, such as *Clostridium* and *Staphylococcus*, are most commonly found in meat, vegetables, dairy products and fruits [4]. In comparison with other foodstuffs, rice and farinaceous foods are significantly more contaminated by *B. cereus* than by other enterotoxin-producing pathogens [50]. By improperly preparing and storing these types of foods, *B. cereus* can become established, grow and produce emetic toxins, which are resistant to subsequent processing [51]. Since emetic *B. cereus* strains do not hydrolyze starch, contaminated starchy foods will taste, smell and look the same as normal foods, except that the rice is sometimes slightly sticky and tastes poor, even if they contain a high number of colonies [52]. The consumption of contaminated foods may result in food poisoning. However, the presence of *B. cereus* in foods is not necessarily associated with disease development; this may occur when the amount of toxin accumulation reaches the emetic dose of vomiting, or the dose levels are sufficiently high to damage organs. Detailed information is included in Section 3 and Section 4.

It is worth noting that the presence of *ces* genes, which regulate cereulide production, cannot guarantee cereulide production, since cereulide synthesis is also determined by multiple other factors, including the food matrix, temperature, pH, water activity (*a_w_*), oxygen level, etc. [53]. A detailed discussion of the factors influencing cereulide production is presented in the subsequent sections.

## 3. Toxicological Profile of Cereulide

### 3.1. Characterization of Cereulide

Cereulide is an emetic *B. cereus* product with the structure of a cyclic and lipophilic dodecadepsipeptide containing three repetitions of four amino acids, D-Oxy-Leu—D-Ala—L-Oxy-Val—L-Val, which resembles valinomycin, a well-known antibiotic formed by *Streptomyces* with the sequence D-Oxy-Hyi—D-Val—L-Oxy-Lac—L-Val (Figure 1a) [54]. In natural niches, cereulide has a potassium-scavenging ability in potassium-poor environments, which provides an adaptive advantage to emetic strains [55,56,57]. It also plays a role in antifungal activity by protecting the mutualistic plant tuber against fungal attraction [58]. The biosynthesis of cereulide is directed by the non-ribosomal peptide synthetase (NRPS) encoded in the *ces* gene clusters, including the *cesA, B, C, D, H, P* and *T* genes, located on a 270 kb megaplasmid known as pCER270 (or pCERE01) (Figure 1b) [59]. The *cesA* and *cesB* genes are responsible for generating the D-Oxy-Leu—D-Ala and L-Oxy-Val—L-Val fragments, respectively, and assembling the two monomers in the peptide chain [60]; the *cesP* gene encodes a 4′-phosphopantetheinyl transferase, which primes the NRPS; the *cesT* gene codes for a putative type II thioesterase, which eliminates misprimed monomers; the *cesC* and *cesD* genes encode for a putative ABC transporter, which is a vital component that acts to restrict the *cesA* and *B* synthase to the cell membrane, playing an important role in peptide assembly [61]; and *cesH* regulates cereulide formation at the transcriptional level [35]. In addition to *cesH*, NRPS cereulide synthesis is also controlled by the AbrB, Spo0A and CodY regulators at the transcriptional level (Figure 1b) [62]. Spo0A inhibits AbrB transcription under low phosphorylation levels, which promotes cereulide formation [62,63]. CodY inhibits cereulide production, as shown in a CodY deletion mutation study where mutated *B. cereus* displayed an upregulation of *cesA* and *cesB* mRNA levels with 60- and 34-fold enhancement, respectively [64].

The initial expression of cereulide occurs in the mid-exponential phase of *B. cereus* growth, and accumulation continues until the stationary growth phase [65,66]. When food is contaminated by cereulide, emetic reactions, including nausea, vomiting and malaise, appear to a generally moderate extent within 0.5–5 h. However, in certain situations, severe effects, including the dysfunction or failure of different organs (liver, intestines, pancreatic islet, etc.), have been observed [24,28].

### 3.2. Absorption, Distribution and Excretion of Cereulide

Cereulide is absorbed into the blood across the intestinal tract and distributed throughout the whole body [23,67]. In the stomach and small intestine, cereulide can bind to the 5-HT3 receptor, resulting in the suppression of mitochondrial activity via the fatty acid oxidation pathway and the subsequent activation of the afferent vagus nerves, leading to an induced emesis mechanism [31,68]. Within the body, one study showed that 48 h after orally dosing pigs with 150 µg/kg cereulide, significantly higher concentrations of cereulide were present in the large intestine (up to 140 ng/g) compared to the small intestine (max. 1 ng/g) [23]. A portion of cereulide entered the bloodstream and was distributed to the spleen, liver, brain and fatty tissues, while some was also directly excreted via the fecal route within 48 h. Moreover, detectable levels of cereulide were more frequently found in fecal samples rather than in blood and urine samples, which may point to the potential risk of false negative results during diagnosis if only some samples, such as serum and urine, are tested [23]. In addition, no reports on the metabolism of cereulide have been published thus far. The excretion of cereulide via urine or feces might be the main pathway of detoxification.

### 3.3. Mode of Action of Cereulide Toxicity

The mode of action underlying the toxicity of cereulide proceeds via the disturbance of the mitochondrial transmembrane potential. K^+^ is fluxed from the outer membrane to the negatively charged inner membrane, leading to the destruction of the electrochemical gradient and causing depolarization, upon which mitochondrial swelling and a lack of ATP driving force are observed. This subsequently impairs the respiration function [68,69,70,71]. Considerable research has shown that cereulide acts as an ionophore, inserting itself into the membrane and subsequently acting as a potassium carrier [57,71,72]. Briefly, cereulide forms an ion channel, which facilitates its diffusion through the hydrophobic interior of cell membranes, maintaining a hydrophilic cavity inside the cyclic molecular structure, which allows the K^+^ to pass through [71,72]. The structure of the inner cyclic dodecadepsipeptide, containing different amino acids and hydroxy acid residues, facilitates easy complex formation with K^+^, resulting in the binding of K^+^ with a slow release into the mitochondrial matrix [57]. After its dissociation from K^+^, free cereulide diffuses back to the cytosol to prepare for the next transport. In this way, the continuous uptake of K^+^ by mitochondria could continue as long as the gradient of the membrane potential exists [57].

### 3.4. Adverse Effects in Different Organs and Body Systems

Cereulide has been reported to cause multi-organ failure, such as liver failure, encephalopathy, pancreatic lysis, acute kidney injury, the necrosis of colon mucosa and mixed intestinal flora [73,74]. Table 1 summarizes the representative adverse effects at the level of organs or body systems, and their potential mechanisms induced by cereulide are also included. The intestine, pancreatic islet, liver, brain (nervous system) and immune system are the target organs, which have received the most attention in previous research [23,24,25,26,27,28,29,30]. As cereulide is absorbed in the intestine, the gastrointestinal tract is the first place where human exposure occurs [25]. In most case studies, cereulide has been shown to cause illness with vomiting and abdominal pain. In one case study, necroses of the colon mucosa and submucosa were observed in a young adult upon the ingestion of pasta containing cereulide [73]. Due to the possibility of cereulide attracting and affecting Caco-2 cells (epithelial cells isolated from the colon), it can lead to mitochondrial dysfunction, the inhibition of intestinal cell proliferation, the disruption of intestinal barrier function and intestinal inflammation [12,75,76]. This may cause necrosis as a result. A detailed description of the effects of cereulide on intestinal cells, including Caco-2 cells and HT-29 cells, has been provided in in vitro studies. Upon chronic exposure to a low dose of cereulide below 1 ng/mL, a reduction in non-mitochondrial respiration, the basal state of respiration, maximal respiration, high-density lipoprotein (HDL) secretion and ATP production have been observed [24,25,27]. The impairment of ATP synthesis impedes the spare respiratory capacity, which facilitates the cells’ response to stress with the additional demand from ATP. Furthermore, the inhibition of intestinal cell proliferation and the disturbance of intestinal barrier function through the downregulation of the intestinal function genes *Occludin*, *Claudin* and *Tff3* have also been observed [25]. In a mouse study, cereulide was shown to play a role in intestinal inflammation, cytokine production and cell apoptosis, which were induced by the activation of endoplasmic reticulum (ER) stress via the *IRE1/XBP1/CHOP* pathway [25]. In addition, exposure to cereulide resulted in the modification of gut microbiota composition by reducing *Lactobacillus*, a gut biomarker of intestinal health that primarily contributes to the production of butyrate. The reduction in butyrate further inhibited the production of 5-HT and induced inflammatory cytokine production, leading to inflammation in the intestines. Moreover, reduced butyrate also decreased food intake in mice [25].

In pancreatic islets, as beta cells rely on mitochondria for aerobic glycolysis and ATP synthesis, cereulide has significant effects on beta cell function and survival [29]. It has been reported that a dose-dependent reduction in glucose-stimulated insulin secretion was observed in MIN6 cells and mouse pancreatic islets when both were exposed to cereulide in a range of concentrations from 0 to 0.15 ng/mL for 24 h. Insulin secretion was completely absent with exposure to 0.25 ng/mL in MIN6 cells and 0.5 ng/mL in mouse pancreatic islets. Moreover, more than 30% of MIN6 cells died after being treated with 0.25 ng/mL cereulide for 24 h. Apoptosis also occurred in the whole cells of mouse pancreatic islets upon exposure to 0.5 ng/mL of cereulide [29]. Similarly, decreased insulin content and increased cell death were demonstrated in fetal porcine Langerhans islets in culture after 2 days of treatment with 1 ng/mL of cereulide [77], and the appearance of necrosis in MIN6 cells was found upon exposure to 10 ng/mL for 8–24 h [56]. The findings of in vitro studies have also been confirmed in clinical outcomes, where pancreatitis or the lysis of the pancreas has also been observed in patients [73,78]. It is unlikely that beta cells generate ATP properly after exposure to cereulide even at low levels due to the mitochondrial dysfunction caused by the low basal oxygen consumption rate, decrease in respiration and increase in ROS production, indicating that cereulide exerts effects on beta-cell-mediated insulin secretion directly instead of affecting insulin synthesis [29]. Additionally, upregulation of the mRNA levels of death protein 5, p53, Atf4 and CHOP was observed in a dose-dependent manner when cells were challenged with cereulide [29], initiating the apoptosis signal or cell death pathway of mitochondria [79].

Mitochondrial function is also impaired in the liver, as shown in both in vitro and in vivo studies [24,30]. Decleer et al. [24] found that the exposure of HepG2 cells to different concentrations of cereulide for 10 days induced adverse effects in these liver cells. A dramatic reduction in maximum respiration was observed upon exposure to cereulide in concentrations ranging from 0.05 nM to 0.5 nM. Maximum respiration declined to 50% and 2% of the original respiration under 0.25 nM and 0.5 nM cereulide treatments, respectively. Moreover, ATP production only remained at 58%, 34% and 6% of the original level (untreated group) with incremental concentrations of cereulide. The adverse effects of cereulide on the liver were validated by the results reported by Yokoyama et al. [30], who exposed mice to cereulide via intraperitoneal injection at 0 (control), 5, 10, 15 or 20 μg/mouse for 1–4 days. Pathological modifications were mainly noticed in the liver rather than in other organs. Further observation showed that hepatocytes derived from mouse livers were found to degenerate at higher dose levels. As doses increased, inflammation reactions, necrosis, the swelling of hepatocytes, the disturbance of cristae, massive microsteatosis and small fatty droplets subsequently appeared. Death of the mice was observed up to the 25 μg/mouse within few hours [30]. When cereulide-induced food poisoning occurs in the clinic, liver failure is the most frequent phenomenon [73,74,80,81,82]. Patients have presented with liver necrosis [73], fatty degeneration of the liver [74], elevated liver enzymes, severe lactic acidosis [82], etc. However, in these cases, the mechanisms of cereulide-induced liver failure were not clarified. A disruption in mitochondrial function may be involved, but further investigation is necessary. In addition, liver damage caused by cereulide may also result in hypoglycemia, as observed in patients with acute liver failure [74,83].

In addition to its harmful effects on organs, the nervous system and immune system have also been shown to be affected by cereulide exposure [23,25,26]. Cereulide arouses depression in different animal species by inhibiting the production of tryptophan hydroxylase 1 and 2 (TPH-1 and TPH-2), key enzymes for the precursor of serotonin (5-HT) synthesis, which controls depression behavior. Cereulide either hinders TPH-1 expression, leading to a decrease in the formation of 5-HT in the intestine, or crosses the blood–brain barrier, possibly inducing apoptosis in the central nervous system (CNS) and exerting its action on the brain, upon which TPH-2 expression is affected [23,25]. The neurobehavioral symptoms observed in pig models following single-dose exposure or repeated exposure over 7 days include recurrent seizures, shivering, lethargic behavior and convulsions of the entire body. These symptoms are similar to those reported in human food poisoning cases [23]. The influence of cereulide on the immune system involves blocking the function of natural killer (NK) cells, which are considered as the first line of defense [26]. NK cells are cytotoxic lymphocytes, which eliminate abnormal cells by releasing cytokines, such as IFN-γ and TNF-α, and initiating apoptosis signaling [84]. A previous study emphasized that only 1 min of exposure to 20 ng/mL cereulide was required for the loss of the cytotoxic capacity of NK cells; 3 h of exposure to 100–1000 ng/mL of cereulide caused the swelling of mitochondria in NK cells; and the induction of apoptosis was observed following 1 day of exposure. Cereulide also partly inhibited IFN-γ production in IL-12, IL-15 and IL-18-stimulated NK cells [26]. Comparable harmful effects were also found for its structural analog valinomycin, which impairs cytotoxicity and cytokine production, enlarges mitochondria and finally triggers apoptosis in NK cells [26].

In addition, cereulide can induce protein expressions or activities, which are associated with cancer cell phenotypes [27]. For instance, enhanced cathepsin D activity, which is normally found in cancer cells and considered an important factor for tumor growth and metastasis, can be promoted by cereulide as well [85,86]. In addition, cereulide-induced mitochondrial dysfunction will lead to increased lactase production, which is also attributed to mitochondrial oxidative stress in cancer cells [87].

**Table 1 foods-12-00833-t001:** Representative adverse effects caused by cereulide at the organ or system level and their potential mechanisms.

Target Organ/Body System	Cell/Animal Model	Concentration/Dose	Exposure Time	Toxic Effects	Mechanism	References
Intestinal	Caco-2 cells	-	0, 2, 4, 8, 12, 24 h	Inhibited intestinal cell proliferation and disrupted intestinal barrier function.	-	Lin et al. (2021) [25]
	Caco-2 cells	0.05–0.5 nM	10 days	Mitochondrial dysfunction; negative effects on the ability of cells to cope with other stresses.	Decreased non-mitochondrial respiration and ATP-linked respiration, especially in maximal respiration; damaged spare respiratory capacity for additional cellular ATP production; downregulation of intestinal function genes *Occludin*, *Claudin* and *Tff3*.	Decleer et al. (2018) [24]; Rajkovic et al. (2014) [27]
Intestinal	HT-29 cells	0.2–500 nM	24 h	Intestinal inflammation.	Sole activation of IRE1/XBP1 signaling pathway; increased expression of C/EBP homologous protein (CHOP), which promotes cell apoptosis during ER stress.	Lin et al. (2021) [25]
	Mice	50 μg/kg body weight	4 weeks	Alterations in the gut microbiota; impact on the biosynthesis of gut microbiota through short-chain fatty acids.	Slight reduction in the relative abundance of *Lachnospiraceae* and *Lactobacillaceae*; decreased level of butyrate production via decreased expression of the Buk gene.	
Pancreatic islet	MIN6 cellsMouse/rat pancreatic islets	0.05 ng/mL–5 ng/mL	24 h and 72 h	Beta cell apoptosis;impaired glucose-stimulated insulin secretion.	Upregulation of mRNA levels of death protein 5, p53, Atf4 and CHOP; reduction in basal oxygen consumption and increase in ROS and PUMA leading to mitochondrial dysfunction and a reduction in ATP production.	Fonseca et al. (2011) [79]; Hoornstra et al. (2013) [56]; Vangoitsenhoven et al. (2014) [29]; Virtanen et al. (2008) [77]
Liver	HepG2 cells	0.05–0.5 nM	10 days	Mitochondrial dysfunction.	Decreased non-mitochondrial respiration and ATP-linked respiration, especially in maximal respiration.	Decleer et al. (2018) [24]
	Mice	5, 10, 15 or 20 µg/mouse	1–4 days	Liver damage and induced death.	Degeneration and increase in fatty droplets in hepatocytes; swelling and loss of cristae in hepatocyte mitochondria; severe lesions in liver at 20 µg and death.	Yokoyama et al. (1999) [30]
Nervous system	Mice	50 μg/kg body weight	4 weeks	Depression-like behavior.	Induction of low levels of serotonin via the inhibition of tryptophan hydroxylase 1 (Tph-1) expression in colon and tryptophan hydroxylase 2 (Tph-2) in brain.	Lin et al. (2021) [25]
	Pig	10–150 µg cereulide kg/body weight	Single exposure	Transient depressive behavior; recurrent seizures; shivering; lethargic behavior; convulsions of the whole body.	-	Bauer et al. (2018) [23]
		10 µg cereulide kg/body weight	Daily exposure for 7 days	Transient depressive behavior; recurrent seizures.	-	
Immune system	NK cells	0–100 ng/mL	1 min–3 h	Inhibition of NK cell cytotoxicity; swelling of mitochondria; induction of apoptosis; inhibitory effect of IL-12 and IL-15 on IFN g production by NK cells.	Dissipation of inner mitochondrial membrane potential.	Paananen et al. (2002) [26]

### 3.5. Toxicity of Isocereulides

Food samples contaminated with *B. cereus* have also been found to contain various isocereulides, including isocereulides A–G, which are analogous to cereulide [88,89,90,91]. Given that isocereulides have similar structures to cereulide, the mode of action underlying the adverse effects of isocereulides is also similar, i.e., their functioning as ionophores in mitochondria [89]. Although the amount of isocereulides A–G found in food is very limited compared to the levels of cereulide, the cytotoxicity of isocereulide A is eight times higher than that of cereulide in HepG2 cells due to variations in the K^+^ transport properties of the two ionophores [88,89]. The variance in K^+^ transport properties can be further supported by the evolution of bilayer membrane conductance, where all toxins can induce an increase in conductance; specifically, the increased conductance triggered by isocereulide A was almost two-fold greater than that induced by cereulide because of its relatively high hydrophobicity. The greater hydrophobicity of structural homologs enables them to better penetrate the membrane, thus increasing the activity of the membrane ionophore [89].

## 4. Infective Doses of *B. cereus*, Emetic Doses of Cereulide and Guideline Levels of *B. cereus*

The infective dose of *B. cereus* is the minimal concentration of *B. cereus* inducing cereulide production, which is ambiguous, since their relationship cannot be expressed in a simple and straightforward manner. The value most frequently reported is 10^5^ CFU/g of food; however, the whole range of infective doses in foods is broad with seven orders of magnitude, starting from 10^3^ CFU/g of food [92,93]. This wide range for the infective dose is likely due to the wide variation in cereulide-producing *B. cereus* strains, their ability to produce the emetic toxins and the environmental factors influencing cereulide formation.

Thus far, the emetic dose of cereulide required to induce emesis in humans is still unclear. Only results obtained from in vitro or animal studies are used as a reference [94]. The estimated emetic doses of cereulide required to cause vomiting in Suncus murinus were 9.8–12.9 ug/kg body weight (bw) [95]; however, these values dramatically exceed the dose levels evaluated in a human situation. In outbreaks of emetic food poisoning, foods implicated in the outbreak contain 0.01 to 1.28 ug cereulide/g food. In an individual weighing 70 kg, consuming 100 g of food will result in a dose of 0.02–1.83 ug/kg bw. [96]. A comparable result was also presented by the Dutch National Institute for Public Health and the Environment (RIVM), which reported that 1.8 ug/kg bw of cereulide could cause detrimental health consequences [48,97]. Moreover, since variable *B. cereus* strains also produce isocereulides with higher toxic potency, the emetic dose is isocereulide-dependent as well [53].

Although there are numerous uncertainties regarding the infective doses of *B. cereus* and the emetic doses of cereulide, guideline levels aimed at restricting *B. cereus* in food have been promulgated globally (Table 2). From Table 2, it can be seen that these restrictions mainly involve specific foods, which are generally associated with vomiting-type outbreaks, for instance, raw foods, foods containing rice or starch, dairy products and ready-to-eat foods. Among these foods, infant formula is a major concern due to the fragility of its consumers. Stricter criteria are set by different countries for formula compared to other types of foods. At the EU level, *B. cereus* concentrations are restricted to a maximum of 500 CFU/g of food [98]. In Canada, in ten units of samples, only one unit is allowed to contain a level of *B. cereus* between 10^2^ and 10^4^ CFU/g of food. These allowable marginal levels are decreased by a factor of 100 compared to those of other foods [99]. The criteria are even stricter in Korea, where in five units of samples, all samples containing *B. cereus* should have levels below 100 CFU/g [100]. These guidelines show that the restriction of *B. cereus* in infant formula has been taken seriously by governments. However, according to a current study [101], cereulide can be produced in these emetic *B. cereus*-concerned foodstuffs where restrictions on cereulide levels are still absent.

## 5. Factors Influencing Cereulide Production

Numerous studies have shown that multiple factors can contribute to cereulide production by influencing: (1) emetic *B. cereus* growth; (2) *ces* genes; (3) biofilm formation, which may have linked to cereulide generation; (4) cereulide synthesis directly. Although *B. cereus* spores are an important feature, thus far there is a lack of evidence to show the role of sporulation in cereulide production (for review, see Huang et al. [20]). Thus, in this section, a discussion of sporulation is not included. Here, we will focus on extrinsic factors that dramatically impact cereulide synthesis. Systematic studies on the impacts of cereulide synthesis at a genetic (intrinsic) level induced by changes in extrinsic factors as well as the potential relation between cereulide synthesis and biofilm production are largely missing, and many studies overlook the contribution of these factors to toxin production. In the following section, the available information related to factors that can impact cereulide production is summarized.

### 5.1. Temperatures

Temperature, as a crucial parameter for cereulide production, has been well studied thus far. Generally, the main cereulide producers are mesophilic *B.* cereus, belonging to phylogenetic group III, with growth temperatures ranging from 10 to 48 °C [105,106]. Recent studies have reported that some psychrophilic *B. weihenstephanensis* strains listed in phylogenetic group VI, such as MC67, MC118 and BtB2-4, generate cereulide as well [107,108]. It has been observed that the temperatures suitable for cereulide production are not always consistent with the temperatures suitable for strain growth. For instance, cereulide production showed a drastic drop when the temperature was above 40 °C and stopped above 43 °C, at which time mesophilic *B. cereus* grew at a faster rate [109,110]. Therefore, it is inaccurate to predict the risks of cereulide by only taking growth parameters into account. In addition to cereulide, temperature also influences the formation and composition of isocereulides. In one study, the amount of isocereulide A shifted from its highest level of around 9–14% of the level of cereulide at low temperatures (12 and 15 °C) to approximately 5% at 18 and 21 °C. In contrast, isocereulide B production increased from 0.8–1.8% (at 12 and 15 °C) to 10% (between 18 and 27 °C) [110]. This raises a new concern regarding low temperatures, which appear to further the formation of the highly toxic isoform.

In cereulide synthesis, the genetic (intrinsic) factors, for instance, *ces* genes or the cereulide synthesis regulator CodY, are also affected by temperature. In a study by Kranzler et al. [110], it was proposed that temperature played a role in cereulide synthesis on a translational and/or post-translational level rather than on a transcriptional level. As the temperature increased, there were inconsistent variations in the transcription levels of *cesB* and the amount of cereulide produced. No significant changes in the transcription level of *cesB* had occurred across the whole temperature range examined (12 or 15 to 33 °C) in four emetic *B. cereus* strains (F4810/72, F5881, B626 and AC01). However, enhanced production of cereulide was observed at 15–18, 30–33 or 30–37 °C depending on the types of strains. These different performances illustrate that temperature has limited effects on cereulide synthesis at a transcriptional level. Comparable results were reported at the *ces* translation level between *cesB* translation and cereulide generation below 40 °C. As the temperature increased above 40 °C, distinct patterns were observed, where the level of cereulide formed was still high, but *cesB* translation decreased significantly. These data revealed that in addition to translation, post-translational regulation mechanisms may also be involved in cereulide production [59,110].

In general, temperature may raise concerns regarding cereulide production, especially for mesophilic *B. cereus* at lower temperatures. It has been reported that the greatest quantities of cereulide were formed by *B. cereus* strains F3744/75 and F2427/76 at 12 and 15 °C rather than at 30 °C or 37 °C, indicating that significant toxin levels can be observed even at a moderate temperature (12 °C or 15 °C) [109]. Furthermore, low temperature plays a role in isocereulide formation and supports the switching of toxin composition toward isoforms with higher toxicity.

### 5.2. pH and a_w_

The effects of pH and *a_w_* on the growth of *B. cereus* have been studied widely. In general, the growth limits for *B. cereus* in terms of pH and *a_w_* values are defined as pH 4.5–9.5 and *a_w_* 0.91–0.99, respectively [16,34,111]. Since these extrinsic factors influencing growth are highly strain and media dependent [112], the range of pH and *a_w_* for growth limits is broad. For pH, the lowest pH value of 5 was proposed by the International Commission for the Microbiological Specifications for Foods (ICMSF) [113]; however, different *B. cereus* strains have different acidity tolerances. For instance, compared to psychrotrophic strains, mesophilic strains are more resistant to acidic conditions [114]. In previous research, the decimal reduction time (D-value) obtained from mesophilic strains was 7.5 min, which was two times longer than that of psychrotrophic strains at pH = 3.5 under 37 °C [115,116]. This implies that potential risks may arise from surviving mesophilic *B. cereus* upon insufficient heating time in an acidic environment. The minimal *a_w_* for emetic *B. cereus* F4810/72 strains, which are tightly related to foodborne outbreaks derived from rice dishes, is 0.941 [117]. Normally, the *a_w_* is adjusted by solute addition, for example, sodium chloride (NaCl) or potassium chloride (KCl). It should be noted that the addition of salt can also contribute to the growth limits of *B. cereus*. The observed maximal salt concentration for *B. cereus* growth is up to 10% [117].

The impacts of pH and *a_w_* on cereulide production and *ces* genes have been reported in some studies [65,66,118]. In one study, slower cereulide production was found in food samples with a lower pH [66]. For instance, vinegar, mayonnaise and ketchup—which contain acetic acid and therefore reduce the pH of the food matrix—inhibited *B. cereus* growth and emetic toxin production. Cereulide production levels were even below the detection limits in the foods [65]. Moreover, F4810/72 strains produced more cereulide in foods at pH 6–7 (neutral), such as béarnaise sauce, liver sausage and cooked rice, than in other conditions, such as Camembert cheese (pH = 7.9) and quark dessert with vanilla flavor (pH = 5.1) [118]. Lower cereulide generation was also found in agar with a lower pH. Psychrotrophic strains (MC67 and BtB2-4) were found to generate decreased cereulide in PCA agar at decreasing pH levels, as follows: pH 7.0 > pH 6.0 > pH 5.4 [119]. However, the effects of pH on *ces* gene levels or the cereulide synthesis process have not yet been identified. 

Studies investigating the impacts of *a_w_* on cereulide synthesis are limited. *B. cereus* is expected to be found in most foods with a pH value higher than 4.8, and dramatically high cereulide production normally occurs in starchy foods with high *a_w_* values [31,92]. For example, bakery products with an *a_w_* > 0.953 and a pH > 5.6 are beneficial to cereulide production. In one study, the greatest cereulide accumulation was found in a rice pastry with an *a_w_* value of 0.982 and a pH of 6.55, followed by a meat pastry filling with an *a_w_* value of 0.988 and a pH of 6.20. In contrast, no detectable increase in cereulide was found in jam rolls or muffins due to their low *a_w_* values of 0.801 and 0.820, respectively [120]. 

Because *a_w_* is altered by the presence of solutes, the impact of solutes on *B. cereus* growth and its capacity for cereulide production cannot be neglected. A study conducted by Biesta-Peters et al. [121] found that the higher the concentration of NaCl (0.3 M and 0.7 M), the greater the reduction in the maximum specific growth rate (μmax) and the longer the delay in lag phase (λ) for *B. cereus*. The addition of NaCl or KCl was also shown to prolong the onset of cereulide production and contributed to the overall level of cereulide formation. A 5 h delay in the onset of cereulide production was observed when media were supplemented with 0.7 M NaCl or KCl as opposed to standard culture media. Moreover, cereulide production decreased by approximately 10-fold upon the addition of 0.3 M NaCl to the media as opposed to normal culture media [121].

In addition, the impact of added salts on *ces* genes expression and consequently on cereulide generation was shown in a study by Dommel et al. [59]. Emetic *B. cereus* was cultured in media supplemented with 0.43 M NaCl. A four-fold reduction in *cesA* transcription and a more than 50% decrease in toxin titer were observed compared to the samples without influences on *B. cereus* growth. As the concentration of NaCl increased to 0.85 or 1.28 M, the inhibition of cell growth was observed, which subsequently affected *cesA* expression and cereulide production. This study indicated that the inhibition of cereulide production may occur in moderately salty foods, even in the absence of an impact on growth. Thus, *ces* expression and cereulide levels cannot be predicted solely based on cell growth or cell numbers.

### 5.3. O_2_/CO_2_/Modified Atmosphere Packaging (MAP)

*B. cereus* strains prefer to grow under aerobic conditions, although they can still survive in facultative anaerobic environments or even through anaerobiosis [53,122]. The absence of oxygen will reduce the growth rate and cell yields of *B. cereus* [123]. According to one study, seven emetic *B. cereus* strains produced mean viable counts of 7.3–7.4 log CFU/mL cells under anaerobic conditions; these counts were substantially lower than those observed under aerobic or microaerobic conditions (7.8–8.6 log CFU/mL), indicating that all strains grew less abundantly in anaerobic conditions [124]. In addition, the replacement of O_2_ with CO_2_ inhibited the maximum specific growth rate of psychotropic *B. cereus* strains. A negative linear relationship was obtained between CO_2_ concentrations and the maximum specific growth rate of psychotropic *B. cereus* [125]. Furthermore, CO_2_ had an effect on the maximum population densities. A reduction in maximum population was recorded at 20% CO_2_, while no growth was observed under 50% CO_2_ conditions. It is worth noting that *B. cereus* can grow aerobically or anaerobically; however, the responses to various stresses in aerobic or microaerobic/anaerobic conditions are different [123]. For example, *B. cereus* grown aerobically is less resistant to heat and acidic conditions than strains grown under anaerobic or microaerobic conditions. Under aerobic conditions with abundant oxygen, excess radicals, such as hydroxyl and/or peroxynitrite, can be formed when cells are exposed to heat or acid and initiate the suicide response. Alternatively, cells are less sensitive to heat and acid in microaerobic or anaerobic conditions, as they are restricted from forming radicals due to the trace amounts or absence of oxygen. Thus, *B. cereus* can survive after heat or acid treatments in a limited oxygen environment, such as modified atmosphere packaging (MAP) and the fulfillment pipelines, which are frequently present in the food industry. It is important to take bacterial behavior, in particular the increased resistance to a certain stress under low oxygen availability, into account, as these conditions apply to the food industry [123].

Low oxygen levels have a negative impact on cereulide production [47,124,126,127,128]. Cereulide production is severely impaired by reduced atmospheric oxygen and completely inhibited by anaerobic conditions with less than 2% O_2_ [126,127]. Thorsen et al. [128] reported that atmospheric oxygen with nitrogen adjusted to 0–2% O_2_ and 20% CO_2_ in N_2_ inhibited cereulide formation in psychotropic *B. weihenstephanensis* strains. This implies that the reduced oxygen concentrations and limited oxygen transfer rate associated with packaging materials contribute to the prevention of potential risks related to emetic toxins from *B. cereus*. The influence of oxygen or carbon dioxide on *ces* genes is still largely unexplored. It is possible that the presence of oxygen in the environment acts more or less as a CodY regulator, resulting in a negative correlation between oxygen levels and cereulide formation [35].

Currently, MAP has been shown to have a protective role, particularly in extending the storage life of goods [126] by changing the composition of gases in contact with food, such as oxygen, carbon dioxide and nitrogen [129]. MAP is also applied widely in the food industry. A previous study investigated three emetic *B. cereus* strains, B116, B203 and F4810/72, which were incubated either with broth or with beans and rice under different gas component conditions for 4 days [126]. Cereulide production was inhibited by at least 50-fold under more than 90% N_2_ atmospheric conditions compared to normal air conditions in broth, and a 10-fold reduction was observed compared to foods. Since some amino acids in rice, such as L-leucine and L-valine, can stimulate cereulide production, anoxic storage is effective in preventing cereulide formation even when these amino acids have been provided. Thus, by using packaging with a modified atmosphere that contains limited oxygen, the risk of cereulide formation during food preservation can be reduced [126].

In addition, MAP can be considered as a critical factor to ensure a sufficiently low level of cereulide produced in sealed packages prior to opening. Thorsen et al. [128] reported that sealed meat packages with the lowest oxygen availability (0% O_2_/20% CO_2_) and oxygen transfer rate (1.3 mL/m^2^) offered the safest choice, with 0.004–0.005 ug cereulide/g production during storage before the opening of the packages at room temperature (20 °C). The initial levels of cereulide production were sufficiently low in the sealed package, leading to a lower risk of toxin formation during storage [128].

### 5.4. Food Matrices/Media/Supplements

Food matrices are important for *B. cereus* growth and cereulide formation. Extensive research has outlined the general characterization of cereulide formation in different types of foods, with the accumulated amount in decreasing order, as follows: carbohydrate-rich foods > proteinaceous foods > fat-enriched foods or vegetables [53,118]. According to Messelhäusser et al. [47], foods can be classified into three main categories, including low risk, risk and high risk, based on real-time monitoring of cereulide production in different foods. The foods in different ranks classified by Messelhäusser et al. [47] correspond to the foods ranked by order of accumulated amount of cereulide, as observed via thorough research. A classification of food matrices could provide information for the food industry to target specific matrices to prevent cereulide toxin production.

In addition to food matrices, different culture media or agars have also been shown to have an influence on cereulide production. For instance, *B. cereus* 5964a and NS117 strains produced higher levels of cereulide in tryptone soy agar (TSA) than in potato dextrose agar, milk agar and nutrient agar at 30 °C [127]. A similar performance was also observed upon the culture of NS58, F4810/72 and NC7401 strains on TSA, blood agar and skimmed milk and raw milk agar. High levels of cereulide synthesis were produced on TSA and blood agar, 5–10 times greater than those on skimmed milk and raw milk agar [130]. The authors indicated that the nutrient composition of culture media may play a main role in cereulide generation. Compared to the high concentrations of K^+^ present in skimmed milk and raw milk agar (35–42 mM), the low concentrations of K^+^ in TSA and blood agar (4–6 mM) stimulated cereulide production, which maintained *B. cereus* K^+^ homeostasis in potassium-deficient environments by facilitating the transport of K^+^ into the cells [55]. It is not surprising that cereulide production is promoted in low-potassium environments. In addition, free amino acids, normally used as food additives or supplements in media, have also been shown to have upregulating or inhibitory effects on cereulide production. Apetroaie-Constantin et al. [131] found a significant correlation between [Na^+^], [K^+^]:[Na^+^], glycine content, the ingredients present in culture media and cereulide generation. High contents of glycine with constant values for [Na^+^] and [K^+^]: [Na^+^] promoted cereulide production. As cereulide is a specific K^+^ ionosphere, and glycine regulates the physiological responses linked to K^+^ homeostasis, these characteristics may explain the increased cereulide formation related to changes in glycine content. With the exception of glycine, the addition of L-valine and L-leucine resulted in the upregulation of cereulide production both in media and in rice and beans, with 4–10-fold enhancements [126]. In contrast, the addition of leucine, isoleucine or glutamic acid reduced the production of the toxin [132]. These findings indicate the potential safety issues associated with additional free amino acids in foods due to their ability to support cereulide accumulation.

Moreover, food additives, such as long-chain polyphosphates (polyPs) [133,134], are also used to prevent cereulide formation in foods. PolyPs were shown to efficiently inhibit cereulide synthesis by downregulating *ces* genes by 3–4-fold at the early stage of the toxin formation process. PolyPs induced a reduction in cereulide production without influencing cellular integrity and growth at low concentrations. The presence of exogenous polyPs may disturb the intracellular balance of the polyP/Pi ratio, subsequently interrupting phosphorelay signal transduction pathways and polyP regulation in *B. cereus*. This process would further influence the Spo0A regulator, which plays a role in the activation of cereulide synthesis [135].

### 5.5. Nutrition Availability

Nutrition availability in the environment is linked to the internal genetic networks of *B. cereus,* which control cereulide production. When nutritional sources in the environment are limited, the CodY regulator would dissociate with GTP, allowing the transition from the exponential phase to the stationary phase [136]. This is also the moment where cereulide starts to be produced. Another indicator of a lack of nutrition is the formation of branched-chain amino acids (BCAAs) [137]. It was reported that *B. cereus* AH187 fed with additional BCAAs had increased CodY effector pool levels, leading to a dramatic decline in cereulide production [64].

### 5.6. Biofilm

The potential association between biofilms and cereulide in the food industry raises food safety concerns due to the fact that biofilms are a source of contamination during food manufacture and processing [138]. It has been found that biofilms have a potential intrinsic link with cereulide production. Two regulators, CodY and AbrB, not only regulate cereulide production but also play a role in the growth of biofilms [36,139]. The negative effects mediated by these regulators were found for both biofilm and cereulide formation. Thus, it is possible that the upregulation of cereulide synthesis may occur along with biofilm development when these factors are downregulated. Moreover, embedded *B. cereus* cells in biofilms (biofilm cells) are capable of releasing metabolites and toxins inside the biofilm [36]. A recent study reported that cereulide toxins attached to the biofilm cells rather than being secreted into the surrounding environment [140]. However, the role of cereulide within the biofilm community is still unclear. It is suspected that toxins, as peptides, might be autoinducers for cell-to-cell communication mediated by quorum sensing (QS) in a microbial community. QS acts by monitoring cell density, which allows communication between bacteria and allows control of specific processes, such as biofilm formation [141]. With increasing cell density, the concentration of QS signals, such as autoinducers, also increases. When the accumulated signals reach a sufficient level, they will activate the maturation and disassembly of the biofilm in a coordinate manner [142]. It is possible that toxins, as the potential autoinducers, may function as a response to high cell density, such as that in the biofilm environment associated with Gram-positive strains [143,144]. It has been reported that toxins are involved in the cell-to-cell communication, which occurs during biofilm development in *S. aureus* [145]. However, further confirmation in *B. cereus* is still needed.

The capacity for biofilm formation on different substances varies in *B. cereus*, and it can further influence cereulide production [140]. In one study, the toxin measured in detached biofilms from stainless steel (SS) wool was almost four-fold higher than that for glass wool (GW). This may be due to the high content of chromium oxide and iron availability on the SS surface, which can promote the growth of *B. cereus* and may also influence cereulide production. In addition, as cereulide has hydrophobic properties and is an ion carrier, hydrophobic surfaces as well as materials with a higher metal content may attach more cereulide. As cereulide attaches to the biofilm matrix instead of being released into the environment [140], it is possible that residual biofilm and cereulide can still be present on surfaces after cleaning procedures. This enables contact between food and the surviving biofilm and cereulide. The effect of substances on cereulide production and accumulation may present a safety concern in the food industry, where SS is commonly used. Furthermore, it is well known that cereulide is highly stable during food processing; however, thus far, there is still a lack of studies investigating the ability of cereulide to attach to food in the absence of *B. cereus*.

## 6. Public Precautions for *B. cereus* and Cereulide

As mentioned previously, it is impossible to eliminate *B. cereus* due to its spore resistance and widespread presence in the environment. In addition, once cereulide is formed, it is unlikely to be destroyed with current food processing techniques. Thus, preventive measures focusing on controlling the growth and toxin production of *B. cereus* are essential to avoid emetic food poisoning [112]. In terms of controlling the growth of *B. cereus* and cereulide production, we propose precautions, which place emphasis on (1) improper temperature usage during preparation, processing and storage, (2) biofilm formation and removal, (3) cross-contamination and (4) regulations regarding the restriction of *B. cereus.* The former three aspects are common problems encountered by food industries, restaurants and consumers. This may be partially due to improper home storage after purchase by customers or food suppliers, as well as non-hygienic operations taking place in the food industry or restaurants. The proposed precautions provide a better insight into prevention strategies at the public level.

As noted above, temperature is a cardinal factor in preventing the growth and toxin production in *B. cereus*. According to emetic *B. cereus* food poisoning cases, temperature abuse is the main contributor to the prevalence of outbreaks. The spores of *B. cereus* are capable of germinating when cooked foods are cooled slowly to room temperature, thus allowing them to grow rapidly in the absence of any competing bacteria killed during cooking [13,146]. Therefore, it is recommended to either eat food immediately after cooking or cool it down as quickly as possible and store it in the refrigerator below 10 °C. Consequently, *B. cereus* growth and cereulide or isocereulide production can be effectively inhibited [147]. It is also suggested that cooked foods should be stored ideally and immediately below 4 °C to prevent the growth of all strains of *B. cereus*, including those that are psychrophiles [148]. Certain countries, including Belgium, require the registration of refrigerator temperatures to guarantee appropriate storage conditions [149,150]. In addition, reheating food is inefficient at killing *B. cereus* or destroying toxins [146,151]. In some cases, even though the precooked foods are reheated before consumption, the population of *B. cereus* is still beyond the safety limits [151].

Another aspect contributing to the survival of *B. cereus* is biofilm production. Resistant spores can attach to surfaces and germinate to produce a biofilm, from which sporulation appears and spreads again [36]. In addition, biofilms play a role as a shelter for cereulide accumulation and can easily induce cross-contamination [36,146]. Thus, the prevention of biofilm formation or the removal of biofilms that are present are crucial steps to reduce potential risks. One way to inhibit biofilm formation is by paying attention to the materials used in the food industry. It is interesting to note that materials that promote biofilm adhesion have an opposite effect on cereulide attachment. According to Xia et al. [152], hydrophilic surfaces, such as glass and SS, are more likely to promote the formation of biofilms than hydrophobic surfaces, whereas hydrophobic surfaces see a greater attachment of cereulide. Considering SS also contains chromium oxide and iron, which promote cereulide attachment, it is suggested that materials coated with antimicrobial agents, such as antimicrobial peptides, essential oils, enzymes, etc., should be used to kill and inhibit *B. cereus* in order to prevent the formation of biofilms [153,154,155]. In addition, regular cleaning and disinfection are currently the main strategies to prevent the formation of biofilms. In the food industry, the rapid and comprehensive cleaning of the contact surface from the initial adhesion of bacteria and timely removal before the biofilm maturation will greatly reduce food safety risks [152].

Furthermore, the prevalence of *B. cereus* can be increased by cross-contamination at product manufacturing levels, for example, from the use of the same equipment for food preparation and raw food materials, cooking utensils, the addition of food additives or ingredients and water supply [146,151,156]. In the inspection of restaurants, cross-contamination is also found due to bad food handling and hygiene conditions, such as uncovered foods stored in the refrigerator, unclean fridge and microwave surfaces, and trash cans with open lids [156]. Although it is difficult to avoid cross-contamination completely, the proper implementation of hazard analysis, critical control points, good agricultural practices and good manufacturing practices, which regulate the sources of cross-contamination, can reduce cross-contamination rate effectively [157].

In terms of regulation, in accordance with the legislation, all operators of food businesses are responsible for producing safe food. An efficient food safety management system based on hazard analysis and critical control points (HACCP) ensures food safety and prevents foodborne pathogens in general. Food safety policies, such as HACCP-based theoretical and practical trainings, are successful in lowering overall microbial prevalence in foodservice establishments, including *Salmonella*, *E. coli* and *Shigella*, which are among the most common causes of foodborne diseases [158]. Furthermore, all food business operators, including primary producers, are legally obliged to implement good hygiene practices (GHP). This regulation requires food business operators to adhere to GHP and HACCP principles in order to ensure compliance with the relevant microbiological criteria. Additionally, education interventions regarding food safety produce positive effects on food handlers’ self-reported knowledge, attitudes and practices (KAP). Improvements in food handlers’ KAP contribute to the reduction in food contamination caused by foodborne pathogens in general [158]. It should also be noted that these food safety management strategies also respond to *B. cereus*. The World Health Organization (WHO) and extensive literature have summarized that incorrect temperature maintenance of foods, improper cleaning of the food production equipment and a lack of staff training are critical factors in reducing *B. cereus* contamination [159]. So far, there are no further official hygiene practices or food safety guidelines for viable cells, spores or toxins of *B. cereus* in foodstuffs. Moreover, there are some limitations in the regulations. Thresholds concerning the absolute amount of cereulide or isocereulide present in foodstuffs are not defined, which should be considered in the future, since in outbreaks, cereulide can still be detected in leftover foods despite the fact that viable *B. cereus* cannot be recovered after the cooking process [146]. Therefore, it is highly recommended that the guidelines be refined by adding additional terms regarding the thresholds of cereulide or isocereulide present in foods based on previous outbreaks. In addition, although some countries have restricted the amount of *B. cereus* in specific food types, which are frequently involved in vomiting-type food poisoning, food matrices categorized as high and medium risk should also receive more attention.

## 7. Conclusions

*B. cereus* strains, with their widespread occurrence in the environment and resistant spores, result in the unavoidable contamination of a wide range of foods with cereulide and isocereulides, which can lead to food poisoning outbreaks. Emetic food poisoning is primarily ascribed to the emetic toxins cereulide and isocereulide, which induce adverse effects by affecting mitochondria in different organs. Currently, it has still not been completely elucidated what the infective doses of *B. cereus* and emetic doses of cereulide are. This is due to the wide variation in cereulide-producing *B. cereus* strains, their different ability to produce emetic toxins and the environmental factors influencing cereulide formation. In fact, the level of cereulide produced is dependent on many influencing factors, including temperature, pH, *a_w_*, oxygen availability, food matrix characteristics and potential biofilm presence.

The impact of emetic *B. cereus* and cereulide on public health cannot be neglected. Precautionary measures aimed at restricting *B. cereus* growth and cereulide production are maintained by the food industry, consumers and regulators; these measures should highlight the importance of temperature control, the material used in the industry, good food handling for avoiding cross-contamination, and regulation refinement, including the definition of thresholds for cereulide and isocereulide. These are comprehensive considerations for the further prevention of the adverse health effects of emetic *B. cereus* and its related toxins. 

## Figures and Tables

**Figure 1 foods-12-00833-f001:**
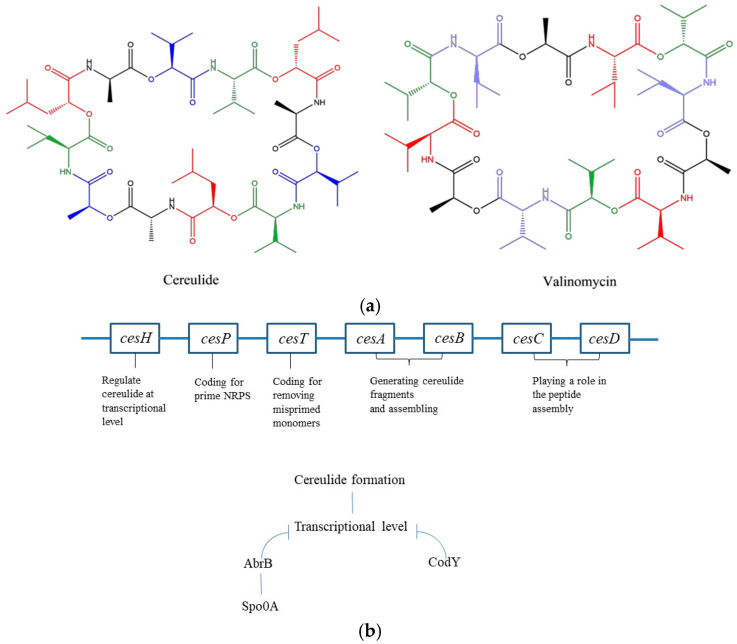
(**a**) Structures of cereulide and valinomycin. For cereulide, the red, black, blue and green fragments represent D-Oxy-Leu, D-Ala, L-Oxy-Val and L-val, respectively. For valinomycin, the red, green, blue and black fragments represent L-val, D-O-Hyi, D-val and L-Oxy-Lac, respectively. (**b**) *ces* genes involved in cereulide formation and regulators at the transcriptional level.

**Table 2 foods-12-00833-t002:** Guideline levels of *B. cereus* in different countries.

Country	Type of Food	Guideline Levels of *B. cereus* (CFU/g or CFU/mL)	References
Australia and New Zealand	Ready-to-eat food	Satisfactory level: <10^2^ Acceptable level: 10^2^–10^3^ Unsafe level: 10^3^–10^4^	NSW Food Authority (2009) [102]
UK	Ready-to-eat food	Satisfactory level: <10^3^ Acceptable level: 10^3^–10^5^ Unsafe level: >10^5^	Public Health England (2009) [103]
EU	Dried infant formula; dried dietary foods for specific medical purposes intended for infants below six months of age	Satisfactory level: <50 Acceptable level: 50–500 Unsafe level: >500	Regulation 1441/2007 [98]
China	Bulk ready-to-eat foods containing rice and flour	Unsafe level: >10^4^	GB 31607-2021 [104]
Canada	Instant infant cereal and powdered infant formula	Sampling parameters ^1^ *n* = 10, c = 1, m = 10^2^, M = 10^4^	Health products and food branch (2008) [99]
	Spices (ready to eat)	*n* = 5, c = 2, m = 10^4^, M = 10^6^	
	Raw organ-derived products and herbal products	*n* = 5, c = 1, m = 10^4^, M = 10^6^	
	Powdered protein, meal replacements and dietary supplements	*n* = 5, c = 1, m = 10^2^, M = 10^4^	
Korea	Infant milk formula, follow-up formula, baby foods for infants or young children and foods for special medical purposes	*n* = 5, c = 0, m = 100 (Except sterilized products)	Food Code [100]
	Ready-to-eat food, fresh-cut products, raw foods, meat and pasteurized or sterilized processed foods	≤10^3^	
	Soy sauces and pastes (except meju), sauce, composite seasoning, kimchi products, salted and fermented seafood products, pickled food products and boiled foods	≤10^4^	

^1^. Sampling parameters: *n*—number of samples or units analyzed; c—maximum allowable number of sample units yielding marginal results, i.e., results between m and M; m—microbiological level that separates good quality from defective, or in a three-class plan, good from marginally acceptable quality; M—microbiological level in a three-class plan that separates marginally acceptable from unacceptable (defective) quality.

## Data Availability

The data presented in this study are available on request from the corresponding author.

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
