# Peer review of "Cereulide and Emetic Bacillus cereus: Characterizations, Impacts and Public Precautions"

_foods, 2023, doi:10.3390/foods12040833_

Round 1
Reviewer 1 Report
This is a well written review article although the topic has been extensively reviewed recently elsewhere.
Minor corrections required particularly on correct species names (italics, capital letters)
Author Response
Thank you for your high evaluation of our manuscript. We really appreciate it. We have corrected the species names with italic and capital letter in the main body of manuscript as well as references. Please see the revised manuscript in attachment.
Reviewer 2 Report
This manuscript has the value of academic and public health management, and the citations of the literature are extremely rich and complete. However, it is recommended to modify or supplement the following information:
It is suggested to change the “title” of B. cereus to Bacillus cereus.
It is suggested to add "B. cereus is easily transmitted by dust and insects to contaminate food, and the carrier rate of food can be as high as 20-70%. Most of the food contaminated by B. cereus will not appear spoilage. Most of the food was normal in appearance, except that the rice was sometimes slightly sticky and tasted poor." or related discussions in this manuscript.
B. cereus is the most commonly related to foodborne illnesses, mainly of vomiting type and diarrhea type. Please add the incubation period information of B. cereus food poisoning, including vomiting type and diarrhea type.
Reviewer 3 Report
A much needed and well written publication. Bacillus cereus is a major problem in the epidemiology of food poisoning. The publication is comprehensive, referring to many aspects. I rate it very highly.
Author Response
Thank you for your high evaluation of our manuscript. We really appreciate it.
Reviewer 4 Report
As requested, I reviewed the manuscript entitled “Cereulide and emetic B. cereus: Characterizations, Impacts and Public Precautions”. This is a review article that provides a comprehensive overview of current knowledge of the natural niches and prevalence of the emetic B. cereus in foods; the toxicological profiles of cereulide; the infective dose of B. cereus; the emetic dose of cereulide; guideline levels for B. cereus; factors that influence cereulide formation; as well as public level precautions for B. cereus and cereulide.
The subject is interesting and relevant in the area of food safety. The work is well organized and comprehensively described, and the bibliographic references used are adequate. Therefore, I believe this article is suitable after minor revisions.
Lines 693-699 - Sentences are unclear. Check punctuation
